# Anomalous Properties of the Dislocation-Free Interface between Si(111) Substrate and 3C-SiC(111) Epitaxial Layer

**DOI:** 10.3390/ma14010078

**Published:** 2020-12-26

**Authors:** Sergey A. Kukushkin, Andrey V. Osipov

**Affiliations:** Institute for Problems in Mechanical Engineering of the Russian Academy of Sciences, 199178 Saint-Petersburg, Russia; andrey.v.osipov@gmail.com

**Keywords:** silicon carbide, epitaxy, SiC/Si interface, ellipsometry, density functional theory

## Abstract

Thin films of single-crystal silicon carbide of cubic polytype with a thickness of 40–100 nm, which were grown from the silicon substrate material by the method of coordinated substitution of atoms by a chemical reaction of silicon with carbon monoxide CO gas, have been studied by spectral ellipsometry in the photon energy range of 0.5–9.3 eV. It has been found that a thin intermediate layer with the dielectric constant corresponding to a semimetal is formed at the 3C-SiC(111)/Si(111) interface. The properties of this interface corresponding to the minimum energy have been calculated using quantum chemistry methods. It has turned out that silicon atoms from the substrate are attracted to the interface located on the side of the silicon carbide (SiC) film. The symmetry group of the entire system corresponds to *P_3_m_1_*. The calculations have shown that Si atoms in silicon carbide at the interface, which are the most distant from the Si atoms of the substrate and do not form a chemical bond with them (there are only 12% of them), provide a sharp peak in the density of electronic states near the Fermi energy. As a result, the interface acquires semimetal properties that fully correspond to the ellipsometry data.

## 1. Introduction

Silicon carbide (SiC) is currently one of the highly demanded wide-bandgap semiconductors [1]. Its mechanical, electrical, and thermophysical properties are close to record levels. The implementation of silicon carbide in well-developed silicon electronics is considered extremely promising [1]. Therefore, methods for the epitaxial growth of SiC on silicon (Si) are of particular relevance. Epitaxy of SiC films on Si by the coordinated substitution of half of the Si atoms with C atoms [2] is a promising method providing a very high quality of SiC films without lattice mismatch dislocations. The term “coordinated” means that the removal of Si atoms from the lattice and the implantation of C atoms in their places due to a chemical reaction occur simultaneously [2]. The oxygen atom here plays the role of a catalyst for the substitution reaction. The transitional state of the reaction (1), corresponding to the maximum energy along the reaction path, is an almost equilateral triangle with Si, C, O atoms at its vertices [3]. After overcoming this energy barrier with a height of 1.2 eV, the SiO molecule leaves the system. Such a mechanism of coordinated substitution preserves the structure of the initial cubic Si lattice, thus ensuring the growth of the cubic 3C-SiC polytype [4].
(1)2Si (crystal) + CO (gas) = SiC (crystal) + SiO (gas) ↑

Numerous studies have shown that the growth of SiC from the material of the Si substrate proceeds in parallel in two ways. The first way is the coordinated substitution of atoms that leads to an epitaxial 3C-SiC film with a thickness of about 100 nm on the sample surface [2,3,4]. The “collapse” of the initial lattice of a Si material with the lattice parameter of 0.543 nm into a cubic SiC lattice with the parameter of 0.435 nm occurs laterally, i.e., in the plane of the substrate [3]. Additional Si atoms required for a uniform filling of the substrate surface diffuse from the depth of the substrate. Besides that, the use of silane SiH_4_ is also a supplement to obtain a uniform ~100 nm thick SiC layer without voids and pits [2]. The second way is a rapid penetration of CO gas into a great depth of the Si substrate. It seems that this mechanism is realized due to dislocation lines and other defects in the initial silicon substrate [5]. As a result, a porous SiC layer is formed inside the substrate under a flat ~100 nm thick SiC film, which is a combination of various SiC polytypes and SiC nanotubes. A thickness of the porous SiC can reach 3–10 µm. According to reaction (1), the volume of the formed SiC is equal to the volume of voids in this layer, since the volume of a Si cell is 2 times the volume of a 3C-SiC cell. A micrograph of a typical 3C-SiC(111)/Si(111) cut is shown in Figure 1.

A number of studies have shown that the ellipsometric spectra of the 3C-SiC(111)/Si(111) samples cannot be described by classical ellipsometric models, in contrast to other similar cases. The reason is that the classical theory uses the Effective Media Approximation (EMA) at the interface between two known materials [6]. In particular, the Bruggeman effective medium approximation [6] gives the following relation, from which the dielectric constant of the mixture $\varepsilon_{eff}$ is determined
(2)$$\underset{i}{\sum }\delta_{i}\frac{\varepsilon_{i}-\varepsilon_{eff}}{\varepsilon_{i}+2\varepsilon_{eff}}=0,$$
where i is the index of a component in the mixture, $\varepsilon_{i}$ is the dielectric constant of the i-th component, and $\delta_{i}$ is its volume fraction. The EMA model (2) usually describes very well the layers at the interface between two media; therefore, this model is fundamental in ellipsometry [6]. According to the EMA, at the interface between two semiconductors, a thin layer is formed with intermediate semiconductor properties.

In this work, it has been revealed for the first time that at the interface between 3C-SiC(111) and Si(111) a thin layer is formed with semimetal properties that cardinally differ from the properties of both SiC and Si and cannot be obtained within the EMA framework. This is caused by the absence of misfit dislocations and by the complex nature of the interaction of two surfaces when the SiC layer attracts individual Si atoms from the substrate. As a result, an interface structure is formed in such a way that 88% of Si atoms at the interface form chemical bonds with the substrate atoms, and 12% of Si atoms at the SiC(111) interface do not form bonds, since they are too far from the Si atoms of the substrate. Such a structure has been described in detail by the methods of quantum chemistry within the framework of the density functional theory. It has been shown that p-electrons of 12% of Si atoms at the SiC(111) interface with dangling bonds provide a sharp peak in the density of electronic states exactly in the Fermi energy region. The conduction band in this case either touches or even goes deep into the valence band by an amount of the order of several hundredths of an eV, which nearly corresponds to the calculation error or is less than it. It has been shown that this theoretical representation is in complete agreement with the experimental results of ellipsometry, according to which the optical properties of the layer at the 3C-SiC(111)/Si(111) interface correspond to the Tauc-Lorentz (TL) parameterization [7] with a bandgap close to zero.

## 2. Experimental Technique and Ellipsometric Analysis

The growth of SiC films was carried out from Si(111) and Si(100) substrates by processing them in a vacuum oven at a temperature of 1280 °C with carbon monoxide gas according to reaction (1). The CO gas pressure and flow rate were 70–350 Pa and 10–20 mL/min, correspondingly. The growth time was in the range of 10–40 min. To obtain uniform SiC layers without pits and cavities, a small amount of silane SiH_4_ (5%) was added to the CO gas. The technological details of the process of epitaxy by the method of substitution of atoms are given in the review [2]. The 3C-SiC films obtained in such a way had a thickness of 40–120 nm when grown on Si(111) and of 30–60 nm when grown on Si(100). It should be noted that at the given temperature, pressure, and gas flow rate, silicon dioxide SiO_2_ is not formed. Analysis of the pressure–flow rate phase diagram for this system showed [8] that silicon dioxide SiO_2_ can be formed at elevated CO pressures due to the reaction $2\mathrm{SiO}\;=\;\mathrm{Si}\;+\;{\mathrm{SiO}}_{2}$. At a given temperature and flow rate the critical CO pressure at which SiO_2_ starts to form is ~1000 Pa.

Ellipsometric analysis of the obtained samples allows them to conditionally divide them into two types [9]. The first type of samples is characterized by the fact that a simple and obvious ellipsometric model very well describes the measured dependences of the pseudodielectric function on the photon energy. Almost all samples grown on Si(100) substrates by the method of coordinated substitution of atoms belong to this type. Figure 2 shows a typical dependence of the pseudodielectric function of the SiC/Si(100) sample on the photon energy, measured with an ultraviolet ellipsometer VUV-WASE (Vacuum Ultraviolet Variable Angle Spectroscopic Ellipsometer, J.A. Woollam Co., Lincoln, OR, USA) with a rotating analyzer in the range of 0.5–9.3 eV. This sample was produced at the CO pressure p_CO_ = 150 Pa, the silane volume fraction of 5%, and the growth time of 20 min. The same figure shows the theoretical ellipsometric spectrum obtained in the framework of the simplest model consisting of only one layer of the film. That is, the lowest layer is the Si substrate containing, within the Bruggeman EMA approximation [6], pores and 3C-SiC in equal volume fractions. On it lies the only layer of this model—the SiC layer containing, in the EMA approximation, voids, and graphite in equal volume fractions. On top of this layer is a standard layer for ellipsometry that describes the roughness of a film, i.e., the material of the film, to which 50% of voids are added in the EMA approximation. One can see from Figure 2 that this simplest ellipsometric model describes without any problems the properties of SiC layers obtained on Si(100) by the method of coordinated substitution of atoms. It should be emphasized that this method is characterized by the presence of some amount of voids and graphite in SiC, which leads to a significant decrease in the height of the peaks in Figure 2 [9]. This feature was explained from a theoretical point of view in [10]. The point is that the method of coordinated substitution of atoms is characterized by the formation of a noticeable amount of silicon vacancies in SiC. The silicon vacancies can be generated both chemically on the SiC surface and due to the effect of ascending diffusion in the bulk of SiC [11]. Further, during the growth, one of the four neighboring C atoms moves to the silicon vacancy, which lowers the energy of the system by 1.5 eV [10]. For this, the carbon atom must overcome an activation barrier with a height of 3.1 eV [10]. Since the typical values of the SiC synthesis temperature are 1200–1300 °C, the thermal fluctuations are quite enough to overcome this barrier. As a result, an almost flat cluster of 4 C atoms is formed with the C—C bond length of 1.57 Å and with the inseparably associated voids with a characteristic diameter of 3.8 Å at the sites of the displaced C atoms [10]. We name such formations in SiC carbon-vacancy structures. Quantum-chemical calculations of the dielectric constant of SiC with the carbon-vacancy structures showed that their contribution is adequately described by the EMA model, in which SiC contains voids and graphite in equal volume fractions [10]. In particular, the ellipsometric analysis of the spectrum of the sample in Figure 2 gives the following results. The volume of the Si substrate contains 1% of pores and SiC, the SiC layer has a thickness of 50 nm and contains 5% of the carbon-vacancy structures, i.e., 5% of pores and 5% of graphite, the roughness is 6 nm.

It is important to emphasize that almost all SiC samples grown on Si by the CVD (Chemical Vapor Deposition) method, regardless of the substrate orientation, belong to the first type of samples, i.e., samples that are described with high accuracy by the simplest ellipsometric model. For example, Figure 3 shows the dependence of the pseudodielectric function on the photon energy for a SiC/Si(111) sample grown by Advanced Epi company by the CVD method with patented technology. This dependence is described by the simplest ellipsometric model, namely, a pure Si substrate with a SiC layer having a thickness of 295 nm and a roughness of 3 nm. It should only be noted that SiC produced by the CVD method is much less transparent than SiC produced by the method of coordinated substitution of atoms.

We refer to the second type of SiC/Si samples as those that cannot be described by any obvious ellipsometric model in principle. Almost all samples grown on Si(111) by the method of coordinated substitution of atoms belong to this type. For example, Figure 4 shows the spectrum of a 3C-SiC/Si(111) sample grown on a Si(111) substrate for 15 min at T = 1280 °C and CO pressure p_CO_ = 100 Pa (the proportion of silane in CO was 5%). Different thicknesses of SiC films grown under the same conditions, but on Si substrates of different orientations (in this case (111) and (100)) are explained by the different hydraulic diameters of channels in the SiC crystal along which CO and SiO gases move towards each other [5]. Figure 4 also shows the best theoretical dependence for this spectrum, obtained in the framework of a rather complex two-layer model with various additives and a classical EMA interface layer between SiC and Si. Moreover, the optical constants of SiC have been chosen as the most optimal, namely in the form of a sum of 3 Tauc-Lorentz oscillators [7], which should have ensured an acceptable agreement between the model and experiment. Nevertheless, as can be seen from Figure 4, no efforts can successfully describe the region of low photon energies up to about 3.3 eV. The experiment gives a rather strong change in the dielectric constant ε in this region, while theory can provide only a slight change in ε (for the given SiC thickness) since SiC is transparent in this region. For the same reason, using the classic EMA interface layer between SiC and Si gives nothing. Note also that ellipsometric analysis does not reveal the presence of silicon dioxide SiO_2_ in the samples of either the first type or the second type.

## 3. Analysis of a 3C-SiC(111)/Si(111) Interface Obtained by the Epitaxial Method of Coordinated Substitution of Atoms

In this work, we are of the opinion that the method of coordinated substitution of atoms forms an interfacial 3C-SiC(111)/Si(111) layer with completely new optical and electrical properties. The growth of SiC on Si(111) by the CVD method does not form such a layer at the interface (obviously, because of lattice mismatch dislocations). Figure 5 shows the X-ray diffraction spectrum of a 3C-SiC(111)/Si(111) sample obtained by the method of coordinated substitution of atoms. One can see that the reflexes correspond only to the <111> direction. There are no other reflexes, which indicates that the epitaxy of 3C-SiC occurs namely along the <111> direction. In addition, there are no polytypes other than the 3C cubic polytype, which is generally characteristic of the growth of SiC on Si. Therefore, typical defects of polytype interfaces are also absent. For the growth on Si(100), reflexes 3C-SiC (111), (110), and (100) are equally present. As shown in [12], this is because the process of substitution of atoms on the (100) face of Si occurs somewhat differently than on the (111) face. In the process of substitution of silicon atoms with carbon atoms, the (100) Si face transforms into a SiC face consisting of an ensemble of facets, the side surfaces of which are covered by the (111) and (110) SiC planes.

It has been found by electron microscopy that lattice misfit dislocations at the 3C-SiC(111)/Si(111) interface are absent, and instead of them, so-called partial dislocations are formed, leading to the formation of stacking faults with interlayers of close-packed hexagonal phases (4H-SiC and 6H-SiC) within the main phase of cubic silicon carbide [2]. Nevertheless, from the shift of the Raman lines of 3C-SiC, it can be concluded that 3C-SiC is compressed (but it is stretched in the CVD method) with the stress of ~0.5 GPa, while characteristic 3C-SiC film thickness is ~100 nm.

To describe the 3C-SiC(111)/Si(111) interface and calculate its properties, we use the methods of quantum chemistry in this work. More specifically, the energy of the 3C-SiC(111)/Si(111) system has been calculated within the framework of the density functional theory and the optimal configuration of atoms corresponding to the minimum energy has been found. For this purpose, the MedeA-VASP (MedeA Vienna Ab Initio Simulation Package) package [13] was utilized. This package helped to analyze a large number of different configurations of the 3C-SiC(111)/Si(111) interfaces obtained by MedeA by the following approach. First, two sets of surface cells are created that are multiples of the original 3C-SiC and Si cells, and a search is performed for the relative orientation with the minimal mismatch between the two surfaces. Then, for all found candidates, geometry optimization is carried out by minimizing the energy in the density functional approximation. Crystal surfaces are not considered in this approach, since they are replaced by periodic boundary conditions. When minimizing the total energy, the VASP package used the PBE (Perdew–Burke–Ernzerhof) functional [14], pseudopotentials, and the plane-wave basis. The cutoff energy of the plane waves was chosen equal to 400 eV. Interfaces between Si(111) and both the Si and C faces of SiC(111) were analyzed. It turned out that the smallest deformation energy of the interface per unit area corresponds to the case when a rhombus of 4 × 4 atoms of the Si(111) surface is simply superimposed on a rhombus of 5 × 5 atoms of the Si face of the SiC(111) surface. All cases with a rotation of one surface relative to the other correspond to higher deformation energy of the interface. An average distance between the Si layer of atoms on the Si(111) surface and the Si layer of atoms on the SiC(111) surface is 2.53 Å, which is 7% larger than the distance between Si atoms in the Si crystal (2.35 Å). The Si(111) substrate almost does not deform SiC, and SiC attracts 1 of 16 Si atoms lying in the bottom of the two layers of the Si(111) double layer, thus modifying the Si subsurface (Figure 6). In fact, this Si atom from the lower layer moves to the upper layer of the double layer of Si(111), forming a bond with a Si atom of SiC that attracted it (Figure 6). The found configuration of atoms at the interface is optimal for the dislocation-free (meaning misfit dislocations) 3C-SiC(111)/Si(111) interface and corresponds to *P_3_m_1_* symmetry. If we consider the interface between Si(111) and the C surface of SiC(111), then SiC will attract not 1 but as many as 3 Si atoms out of 16 (in full accordance with the requirement of *P_3_m_1_* symmetry). Consequently, the deformation energy of the interface will be somewhat higher, but not much. Thus, it follows from the quantum-chemical calculations that the surface of a 3C-SiC(111) film growing on Si(111) by the method of coordinated substitution of atoms should be of the C-type since the Si-type surface is oriented toward silicon. A comparison of the total energy of the entire system for these two cases is beyond the scope of this work since here it is necessary to take into account the real reconstruction of the SiC surface [15].

An interesting feature of the found interface is that its bandgap decreases to almost 0. Calculation of the band structure of this system shown in Figure 6 was carried out using the SCAN functional [16], which is a significant improvement of the usual GGA functionals, in particular, PBE. The calculation has shown that the conduction band enters the valence band by a few hundredths of eV. In other words, the interface between the two semiconductors is a semimetal. Figure 7 shows the dependence of the density of electronic states of the system under study on the energy (the Fermi energy corresponds to 0). One can see that there is a sharp peak in the density of electronic states in the vicinity of the Fermi energy, which provides unusual electrical and optical properties of the system. The analysis has shown that this peak of the density of electronic states is due to the p-electrons of 3 silicon atoms out of 25 in SiC at the interface with Si. In Figure 8 these atoms are shown by arrows, and in Figure 6 they are highlighted in red. Their peculiarity is that they are located as far as possible from the Si atoms of the substrate and do not form chemical bonds with them. In other words, it is these 12% of Si atoms at the interface have one unsaturated bond, which ultimately leads to the semimetallic properties of the 3C-SiC(111)/Si(111) interface.

## 4. Ellipsometric Analysis of the 3C-SiC(111)/Si(111) Samples with an Account of the Semimetal Layer at the Interface

Ellipsometric analysis of the 3C-SiC(111)/Si(111) samples has shown that, at the interface between the SiC and Si semiconductors, a layer appears with unusual optical properties, which cannot be described in the framework of EMA in principle (Figure 4). To describe them theoretically, one has to use a model with oscillators that are in no way related to either SiC or Si. Oscillators of Lorentz, Drude, Gauss give good results. However, the best result for the vast majority of samples is given by the Tauc-Lorentz (TL) model with either a zero bandgap or a very small one (<0.5 eV). Thus, the best universal model describing the optical properties of 3C-SiC(111)/Si(111) samples produced by the method of coordinated substitution of atoms is as follows (Figure 9).

The Si substrate is represented by a mixture with voids and 3C-SiC taken in equal volume fractions. The dielectric constant of the mixture is calculated within the framework of the classical EMA Bruggeman model (2). Equal fractions of silicon carbide and voids arise from the fact that the volume of the initial Si cell is 2 times greater than the volume of the 3C-SiC cell resulted from reaction (1). The first layer on the substrate is a semimetal layer with unknown optical properties, so they are described by the model of one TL oscillator.
(3)$$\varepsilon_{2}\left(E\right)=\{\begin{array}{ll} \frac{1}{E}\;\frac{AE_{0}C{\left(E-E_{g}\right)}^{2}}{{\left(E^{2}-E_{0}^{2}\right)}^{2}+C^{2}E^{2}}, & \;E>E_{g}\\ 0, & \;E\leq E_{g} \end{array}$$

Here *E* is the photon energy, *ε*_2_ is the imaginary part of the dielectric constant (the real part *ε*_1_ is calculated from *ε*_2_ using the Kramers-Kronig relation [6]), *A*, *E*_0_, *C*, *E_g_* are minimization parameters of the model. *E_g_* has the meaning of the bandgap, *E*_0_ is the position of the oscillator peak, A is its amplitude, C is its half-width. If *E_g_* = 0, then expression (3) is simplified, and the number of the minimization parameters of the model is reduced to 3. We emphasize that the thickness of the first layer (i.e., semimetal) *h*_1_ (Figure 9) strongly correlates with the amplitude of oscillator A. Therefore, it is impossible to determine together *h*_1_ and *A* from the experimental data, since a change in *h*_1_ in the range from 0.5 to 5 nm almost does not lead to a change in the error function. In this work, the thickness of the semimetal layer *h*_1_ is assumed to be equal to the average interface roughness determined by microscopic exploration, that is, *h*_1_ = 2 nm.

The second layer is 3C-SiC with a thickness *h*_2_ (Figure 9), containing in equal proportions voids and graphite, i.e., carbon-vacancy structures. Since their concentration is low (from 0 to 8%), it is not necessary to use the Bruggeman model, the simpler Maxwell-Garnett model [6] is sufficient. Finally, on top of the second layer, there is a roughness of thickness *h*_r_ (Figure 9), i.e., the Bruggeman mixture of 50% of the material of the second layer and 50% of voids [6]. For *h*_1_ = 2 nm and *E_g_* = 0, this ellipsometric model contains only 7 minimization parameters. There is practically no correlation between these parameters, so they are uniquely determined by the measured ellipsometric spectrum. Figure 10 shows both experimental and theoretical dependences of the pseudodielectric constant ε of the 3C-SiC(111)/Si(111) sample on the photon energy *E*. One can see that taking into account the interface layer with semimetallic optical properties (3) (at *E_g_* = 0) makes it possible for the first time to adequately describe the experimental ellipsometric data (Figure 10). In particular, for the sample under study, *h*_2_ = 48 nm, *h*_r_ = 5 nm, the volume concentration of SiC and voids in the substrate is 28%, and the concentration of carbon-vacancy structures in SiC is 0.5%. The dielectric constant of the interface layer is shown in Figure 11. We emphasize that for almost all 3C-SiC(111)/Si(111) samples produced by the method of coordinated substitution of atoms, the maximum of the *ε*_2_ function for the interface layer is about 2 eV. Since the half-width of the TL peak is also close to 2 eV, *E_g_* is always very close or equal to 0. The conductivity of the interface layer can be estimated by comparing its dielectric constant with the dielectric constant of the conductor, in particular, in Figure 11, the dashed line indicates *ε*_2_ of a conductor in the Drude model [17] with a resistivity of 4 × 10^−7^ Ohm m and a scattering time of conduction electrons 4 × 10^−1^^6^ s (this is about 2 times worse than that of lead). That is, in the frequency range of an electric field of 700 THz and over, the interface layer conducts current like a bad metal. However, in the region of low frequencies, its conductivity deteriorates significantly and becomes similar to the conductivity of a semiconductor with a zero bandgap [17]. In general, the conductivity of the interface layer is determined by a sharp narrow peak in the density of electronic states near the Fermi energy (Figure 10).

## 5. Conclusions

In summary, in this work, it has been revealed that the ellipsometric spectra of silicon carbide layers grown by the method of coordinated substitution of atoms on the Si(111) surface are fundamentally different from the ellipsometric spectra of similar layers grown either by another method or on a Si surface with a different orientation. The reason was found to be that at the interface of 3C-SiC(111)/Si(111) there is an intermediate layer whose dielectric constant cannot be described within any EMA model. By its appearance, the dielectric constant is mostly similar to that of a semimetal and is well described by the Tauc-Lorentz model with zero or very small (< 0.5 eV) bandgap. The root-mean-square error in the description of ellipsometric spectra decreases by about 3 times by using the TL model as compared to the EMA.

A quantum-chemical model of the 3C-SiC(111)/Si(111) interface has been constructed, which explains the reasons for the appearance of the layer with the properties of a semimetal. It has been shown that, for dislocation-free matching of SiC and Si lattices differing by 20%, the SiC film oriented with its Si surface toward the substrate attracts one of 16 Si atoms in the nearest double atomic layer of the substrate. At that, 22 out of 25 Si atoms form chemical bonds with the Si atoms of the substrate, and 3 out of 25 atoms (i.e., 12%) do not form bonds, since they are too far from the substrate atoms (over 3 Å). The p-electrons of namely these Si atoms in SiC make the main contribution to a narrow and sharp peak of the density of electronic states of 3C-SiC (111)/Si(111), located in the vicinity of the Fermi energy. In other words, the 3C-SiC(111)/Si(111) interface must have unusual electrophysical properties, in particular, it must conduct electric current well. The measured ellipsometric spectra show that the conductivity of the interface is only 2 times worse than that of a metal like lead at electric field frequencies over 700 THz. The conductivity deteriorates at lower frequencies.

## Figures and Tables

**Figure 1 materials-14-00078-f001:**
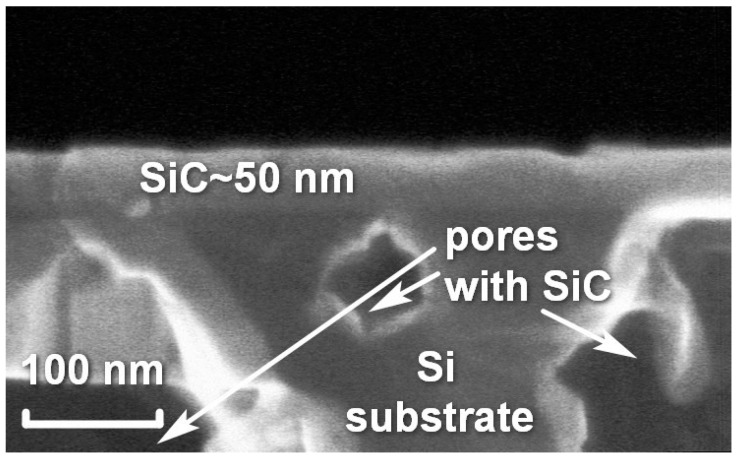
Micrograph of a section of a 3C-SiC(111)/silicon (Si)(111) sample grown by the method of substitution of atoms.

**Figure 2 materials-14-00078-f002:**
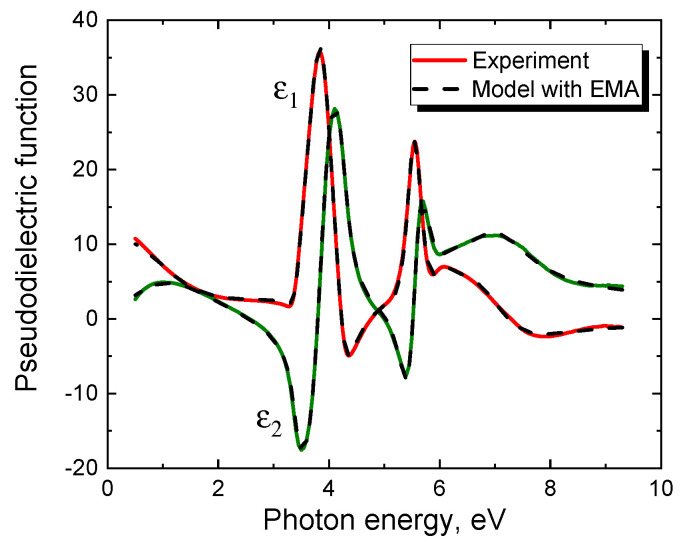
Dependence of the pseudodielectric function (*ε*_1_ is its real part, *ε*_2_ is its imaginary part) of a silicon carbide (SiC)/Si(100) sample on the photon energy, measured in the range of 0.5–9.3 eV by an ultraviolet ellipsometer J.A. Woollam VUV-WASE with a rotating analyzer (solid line). The dashed line is the same relationship calculated using the Effective Media Approximation (EMA) model.

**Figure 3 materials-14-00078-f003:**
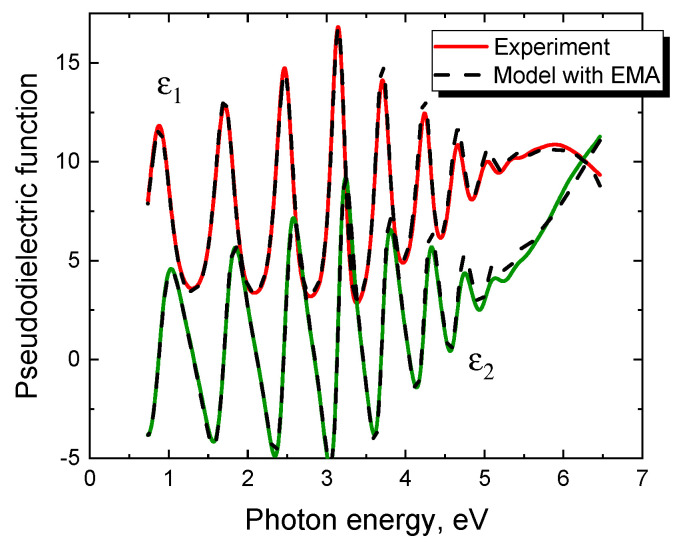
Dependence of the pseudodielectric function (*ε*_1_ is its real part, *ε*_2_ is its imaginary part) on the photon energy of a SiC/Si(111) sample grown by the CVD method by Advanced Epi using patented technology (solid line). The dashed line is the same relationship calculated using the EMA model.

**Figure 4 materials-14-00078-f004:**
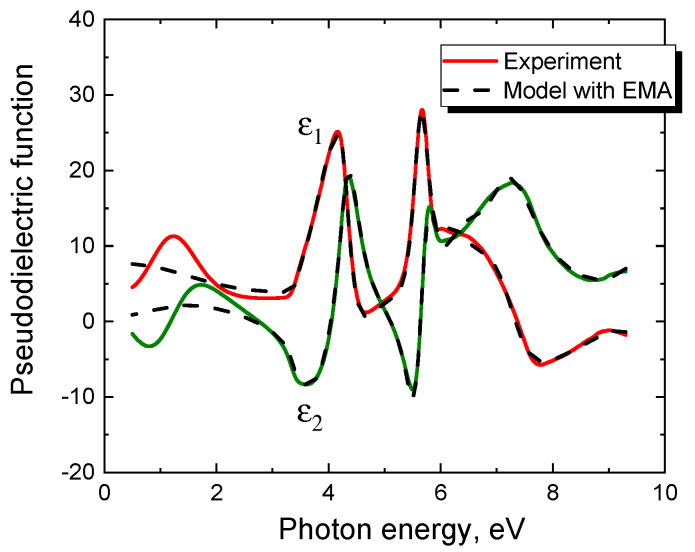
Dependence of the pseudodielectric function (*ε*_1_ is its real part, *ε*_2_ is its imaginary part) on the photon energy of a 3C-SiC/Si(111) sample grown by the method of substitution of atoms on a Si(111) substrate for 15 min at T = 1280 °C and CO pressure p_CO_ = 100 Pa (solid line). The dashed line is the best theoretical dependence for this spectrum, obtained within a complex two-layer model using EMA.

**Figure 5 materials-14-00078-f005:**
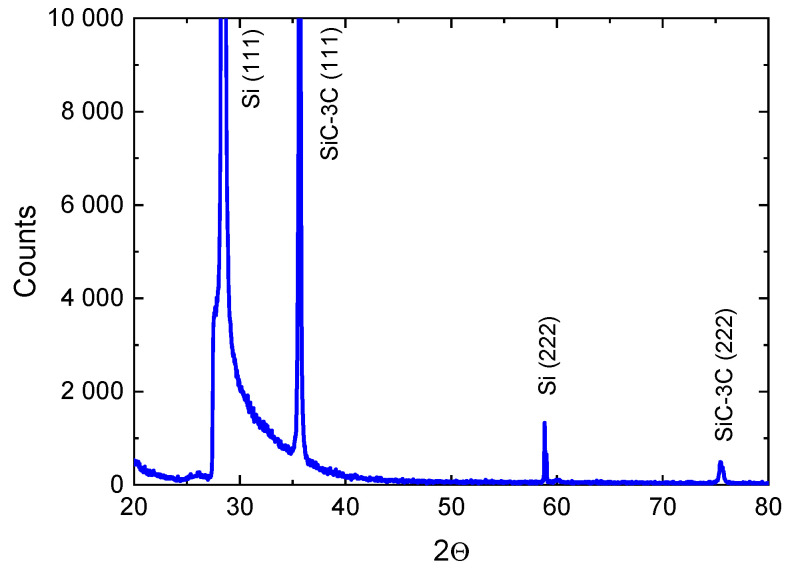
X-ray diffraction spectrum of a 3C-SiC(111)/Si(111) sample grown by the method of coordinated substitution of atoms.

**Figure 6 materials-14-00078-f006:**
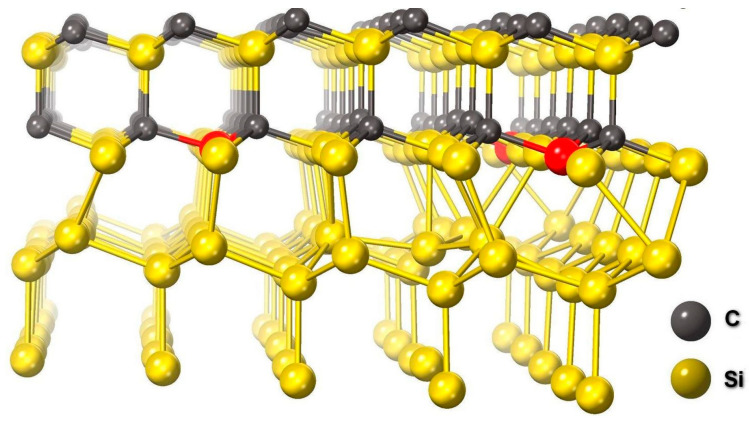
Atomic configuration corresponding to the optimal dislocation-free 3C-SiC(111)/Si(111) interface, calculated by the methods of quantum chemistry in the framework of DFT. The symmetry of the system corresponds to *P_3_m_1_*. The red color highlights those Si atoms at the SiC boundary that do not form bonds with the Si atoms of the substrate since they are too far from those.

**Figure 7 materials-14-00078-f007:**
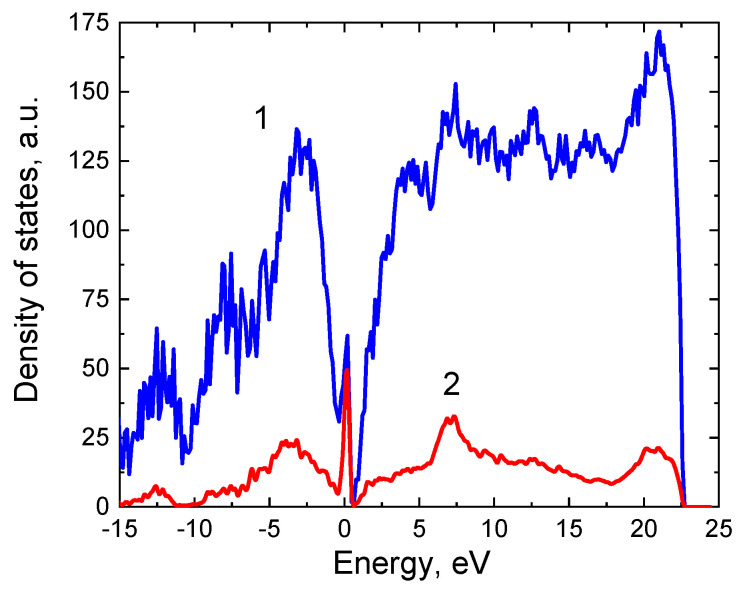
Dependence of the density of electronic states of the system under study on energy (curve 1). The Fermi energy corresponds to 0. Curve 2 is the contribution of p-electrons of those Si atoms at the SiC boundary that do not form bonds with the Si atoms of the substrate (highlighted in red in Figure 6 and by arrows in Figure 8).

**Figure 8 materials-14-00078-f008:**
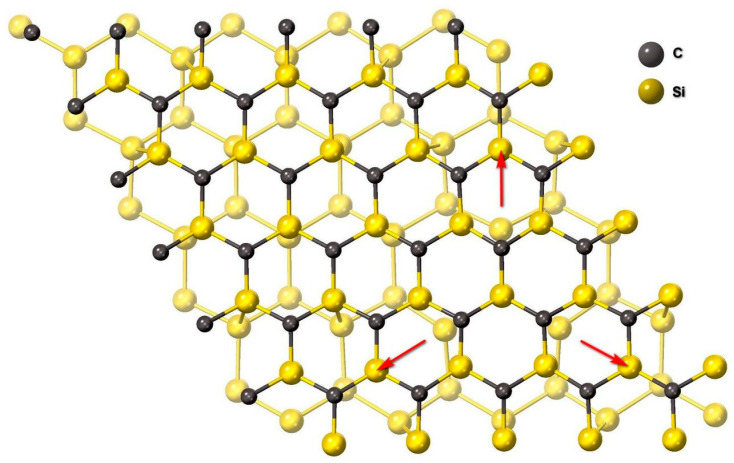
View of the optimal dislocation-free interface 3C-SiC (111)/Si (111) perpendicular to the interface. Arrows show Si atoms at the SiC interface, which do not form bonds with Si atoms of the substrate.

**Figure 9 materials-14-00078-f009:**
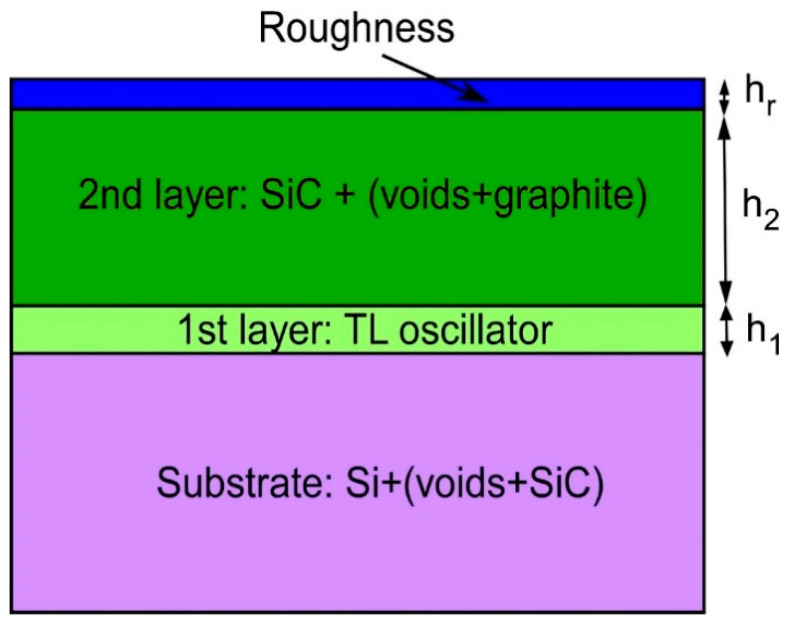
Universal two-layer ellipsometric model describing the optical properties of 3C-SiC(111)/Si(111) samples produced by the method of coordinated substitution of atoms. The optical constants of the intermediate layer separating 3C-SiC and Si are described by the Tauc-Lorentz (TL) oscillator.

**Figure 10 materials-14-00078-f010:**
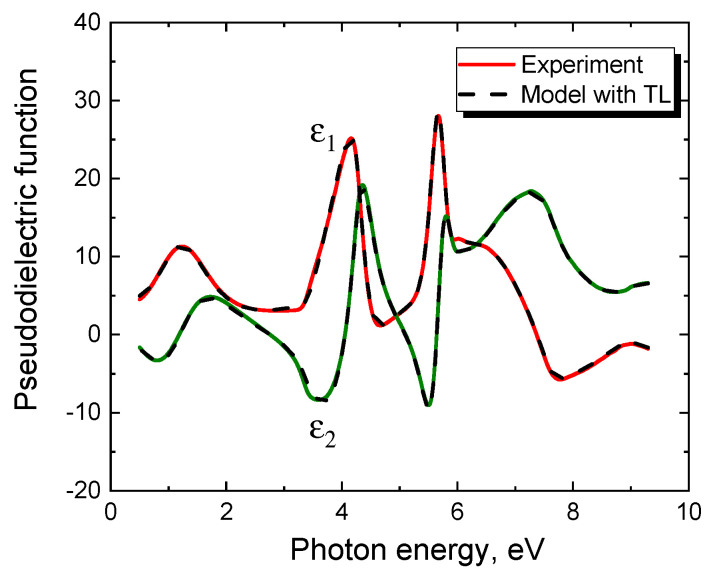
Comparison between the experimental (solid line) and theoretical (dashed line) dependences of the pseudodielectric constant ε on the photon energy of a 3C-SiC(111)/Si(111) sample grown by the method of substitution of atoms on a Si(111) substrate. The theoretical curve was obtained on the basis of an ellipsometric model with a semimetal layer at the interface (Figure 9).

**Figure 11 materials-14-00078-f011:**
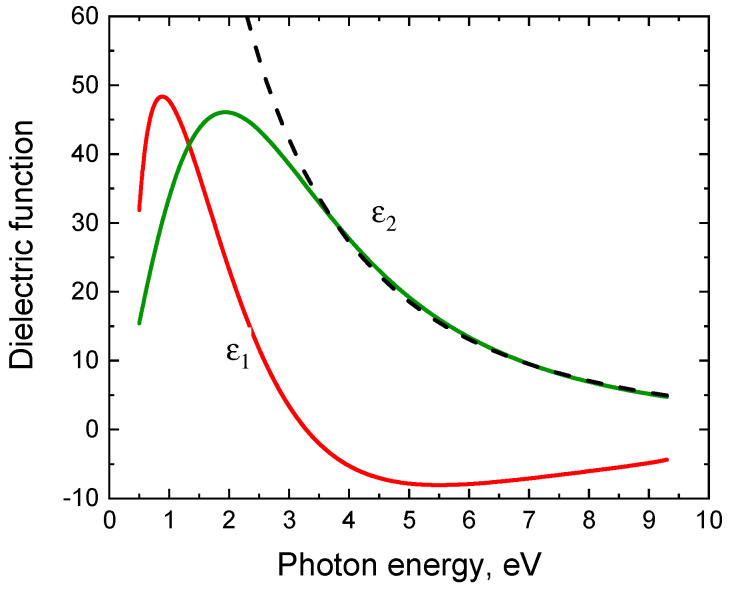
Dependence of the dielectric constant (*ε*_1_ is its real part, *ε*_2_ is its imaginary part) of the interface layer of a semimetal on the photon energy, measured in the range of 0.5–9.3 eV by an ultraviolet ellipsometer J.A. Woollam VUV-VASE with a rotating analyzer on the assumption that the thickness of the semimetal interface layer is *h*_1_ = 2 nm.

## Data Availability

Data is contained within the article.

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
