# Peer review of "Anomalous Properties of the Dislocation-Free Interface between Si(111) Substrate and 3C-SiC(111) Epitaxial Layer"

_materials, 2020, doi:10.3390/ma14010078_

Round 1

Reviewer 1 Report

The paper continues a series of research works made by the authors on the formation of SiC layers on Si substrates by the coordinated substitution of atoms method. In particular, unusual semi-metallic properties of the SiC/Si interface layer have been disclosed and the structure of this layer has been proposed based on the quantum chemistry calculations. The paper is clearly written and convincing. I find it quite suitable for the publication in the Materials journal.

Prior to the publication, however, the following issues have to be clarified and improved:

1. The synthesis method includes reaction of Si atoms from the substrate with oxygen. I would expect formation of some noticeable quantity of SiO2, especially taking into account that CO can penetrate deep into the Si substrate, where from the removal of SiO by-product may become problematic. Could the authors comment on the quantity of formed SiO2 and its influence on the measured ellipsometric spectra?

2. SiC is known to exist in many polytypes, the energy difference between which is small. Therefore, multiple stacking faults forming inclusions of different phases in SiC layers are common. Presence of stacking faults in the SiC layers obtained by the authors is implied by the presence of partial dislocations terminating them. How do the multiple stacking faults and different-polytype inclusions influence on the measured ellipsometric spectra and their interpretation?

3. According to the authors, the four-layer elliprometric model uses 7 minimization parameters, which actually provide certain information about the characteristics of the grown SiC layers. Authors should comment on how unique is the set of parameters found by them.

4. Finally, I believe that the indications of thicknesses h1, h2 and hr should be placed also in Fig. 9 for better clarity.

Author Response

Dear Reviewer 1, thank you very much for a very important comments.

Answer on questions Reviewer 1

  1. Question 1.

The synthesis method includes reaction of Si atoms from the substrate with oxygen. I would expect formation of some noticeable quantity of SiO2, especially taking into account that CO can penetrate deep into the Si substrate, where from the removal of SiO by-product may become problematic. Could the authors comment on the quantity of formed SiO2 and its influence on the measured ellipsometric spectra?

Answer 1

Dear Reviewer, thank you very much for a very important comment.

Earlier, in Ref. [8], we fully calculated all possible chemical reactions that can take place during the chemical interaction of Si with the gas CO. These calculations showed that SiO2 is only reversed if the pressure in the working chambers is equal to or exceeds 1000 Pa. These calculations showed that silicon oxide is formed only if the gas pressure in the working chamber is equal to or exceeds 1000 Pa at the growth temperature. We grow silicon carbide at pressures not exceeding 200 Pa.  We carried out numerous studies of the composition of the layers, in which it was shown that silicon dioxide (SiO2) is not formed in our process. These studies are described in detail in the review [4]. Analysis of ellipsometric spectra also did not reveal the presence of SiO2. We took your comments into account and inserted additional text explaining this on page 3,4, 5 and 6 into the article. We highlighted it in yellow.

We inserted this text into the article on pages 3 and 4

“It should be noted that at the given temperature, pressure, and gas flow rate, silicon dioxide SiO2 is not formed. Analysis of the pressure — flow rate phase diagram for this system showed [8] that silicon dioxide SiO2 can be formed at elevated CO pressures due to the reaction . At a given temperature and flow rate the critical CO pressure at which SiO2 starts to form is .”

We inserted this text into the article on pages 5 and 6

“Note also that ellipsometric analysis does not reveal the presence of silicon dioxide SiO2 in the samples of either the first type or the second type.”

  1. Question 2

SiC is known to exist in many polytypes, the energy difference between which is small. Therefore, multiple stacking faults forming inclusions of different phases in SiC layers are common. Presence of stacking faults in the SiC layers obtained by the authors is implied by the presence of partial dislocations terminating them. How do the multiple stacking faults and different-polytype inclusions influence on the measured ellipsometric spectra and their interpretation?

Answer 2

Dear Reviewer, thank you very much for a very important comment.

SiC films grown under these conditions do not contain any other polytypes except for cubic. The X-ray spectrum shown in Fig. 5 unambiguously testifies to this. We inserted this text into the article on page 6

“Fig. 5 shows the X-ray diffraction spectrum of a 3C-SiC(111)/Si(111) sample obtained by the method of coordinated substitution of atoms. One can see that the reflexes correspond only to the <111> direction. There are no other reflexes, which indicates that the epitaxy of 3C-SiC occurs namely along the <111> direction. In addition, there are no polytypes other than the 3C cubic polytype, which is generally characteristic of the growth of SiC on Si. Therefore, typical defects of polytype interfaces are also absent.”

Question 3

According to the authors, the four-layer elliprometric model uses 7 minimization parameters, which actually provide certain information about the characteristics of the grown SiC layers. Authors should comment on how unique is the set of parameters found by them.

Answer 3

Dear Reviewer, thank you very much for a very important comment.

There is no correlation between these 7 parameters and therefore there is no ambiguity in their definition. We added this phrase to the text of the article on page 9.

“There is practically no correlation between these parameters, so they are uniquely determined by the measured ellipsometric spectrum.”

Question 4

Finally, I believe that the indications of thicknesses h1, h2 and hr should be placed also in Fig. 9 for better clarity.

Answer 4

We indicated these three thicknesses in Fig. 9.

Reviewer 2 Report

The manuscript proposed an anomalous semimetal behaviour at the dislocation-free interface between Si and the SiC epi-layer. 

The reviewer has the following questions:

Why a different thickness is observed depending on the substrate orientation?

"The 3C-SiC films obtained in such a way had a thickness of 40 – 120 nm when grown on Si(111) and of 30 – 60 nm when grown on Si(100)."

How many samples follow the different ellipsometric models proposed?

Number of samples showing the semimetal thin layer?

Would it be possible to see the semimetal thin film by TEM?

Author Response

Dear Reviewer 2, thank you very much for a very important comments.

Reviewer 2

Question 1.

  1. Why a different thickness is observed depending on the substrate orientation? "The 3C-SiC films obtained in such a way had a thickness of 40 – 120 nm when grown on Si(111) and of 30 – 60 nm when grown on Si(100)."

Answer 1

The difference in the thickness of the films grown on substrates Si of different orientations is associated with the difference in the mechanisms of gas CO mass transfer to the depth of these substrates. These mechanisms are described in the ref. [5]. We have included this phrase in the text of our article on page 6.

“Different thicknesses of SiC films grown under the same conditions, but on Si substrates of different orientations (in this case (111) and (100)) are explained by the different hydraulic diameters of channels in the SiC crystal along which CO and SiO gases move towards each other [5].”

  1. How many samples follow the different ellipsometric models proposed?

Answer 2

Over the years, we have measured about two thousand samples.

  1. Number of samples showing the semimetal thin layer?

Answer 3

Of the two thousand samples, about a thousand were obtained on the substrates of  (111) orientation. They all have this property.

  1. Would it be possible to see the semimetal thin film by TEM?

       Answer 4

Silicon atoms in the semi-metal state are indistinguishable from ordinary silicon atoms. In our opinion, with the help of TEM it is not possible to distinguish them, in any case, now.
